# Racialized economic segregation and health outcomes: A systematic review of studies that use the Index of Concentration at the Extremes for race, income, and their interaction

**Anders Larrabee Sonderlund**[1,2]*, **Mia Charifson**[1], **Antoinette Schoenthaler**[3], **Traci Carson**[3], **Natasha J. Williams**[3]

**1** Department of Population Health, NYU Grossman School of Medicine, New York, New York, United States of America, **2** Research Unit of General Practice, Department of Public Health, University of Southern Denmark, Odense, Denmark, **3** Center for Healthful Behavior Change, Institute for Excellence in Health Equity, New York, New York, United States of America

* anders.larrabeesonderlund@nyulangone.org

## Abstract

Extensive research shows that residential segregation has severe health consequences for racial and ethnic minorities. Most research to date has operationalized segregation in terms of either poverty *or* race/ethnicity rather than a synergy of these factors. A novel version of the Index of Concentration at the Extremes (ICE$_{Race-Income}$) specifically assesses racialized economic segregation in terms of spatial concentrations of racial and economic privilege (e.g., wealthy white people) versus disadvantage (e.g., poor Black people) within a given area. This multidimensional measure advances a more comprehensive understanding of residential segregation and its consequences for racial and ethnic minorities. The aim of this paper is to critically review the evidence on the association between ICE$_{Race-Income}$ and health outcomes. We implemented the Preferred Reporting Items for Systematic Reviews and Meta-Analyses guidelines to conduct a rigorous search of academic databases for papers linking ICE$_{Race-Income}$ with health. Twenty articles were included in the review. Studies focused on the association of ICE$_{Race-Income}$ with adverse birth outcomes, cancer, premature and all-cause mortality, and communicable diseases. Most of the evidence indicates a strong association between ICE$_{Race-Income}$ and each health outcome, underscoring income as a key mechanism by which segregation produces health inequality along racial and ethnic lines. Two of the reviewed studies examined racial disparities in comorbidities and health care access as potential explanatory factors underlying this relationship. We discuss our findings in the context of the extant literature on segregation and health and propose new directions for future research and applications of the ICE$_{Race-Income}$ measure.

**Data Availability Statement:** All relevant data are within the manuscript and its Supporting Information files.

**Funding:** The author(s) received no specific funding for this work.

**Competing interests:** The authors have declared that no competing interests exist.

## Introduction

In the United States (U.S.), racial and ethnic minorities fare significantly worse than their white counterparts on nearly all health outcomes, including cancer, cardiometabolic disease, infant mortality, and mental health. Based on a robust and growing body of evidence, structural racism in the U.S. has been implicated as a key determinant of racial and ethnic health inequities. Structural racism is defined as the inherent mechanisms of society that preserve systems of white privilege by perpetuating racism in multiple, mutually reinforcing societal domains [1, 2]. These include, but are not limited to systems of health care, education, housing, employment and economic opportunity, environment, and criminal justice–all of which may interact to impact negatively on racial and ethnic minority health [3, 4]. Racialized residential segregation, in particular, has been identified as one of the most pervasive and persistent mechanisms through which structural racism produces health inequality [5].

Racialized residential segregation in the U.S. is the result of a long history of explicit racist policies (e.g., Jim Crow laws) that continue to be perpetuated by local and federal housing policies (e.g., redlining) and economic practices (e.g., home lending) and enforced by aggressive and violent policing by law enforcement and citizens [2, 6, 7]. These racist structures are designed, in part, to spatially concentrate and disenfranchise racial and ethnic minority populations from mainstream white society, thus restricting access to resources and opportunity for optimal physical and mental health and economic affluence and mobility [2, 8]. A strong body of evidence connects segregation with increased rates of cardiovascular and metabolic diseases, cancer, adverse birth outcomes, obesity, health-risk behavior, and all-cause mortality [9–11]. For example, segregation has been associated with a nearly threefold increase in premature death for Blacks compared to whites [12]. Yet, the underlying processes that underpin the observed association between segregation and health outcomes are understudied and unclear.

Socioeconomic status (SES) represents one pathway through which segregation affects the health of racial and ethnic minorities [13, 14]. Specifically, research highlights the fact that while segregation benefits privileged communities socioeconomically, it simultaneously confers significant disadvantage on segregated racial and ethnic minorities on a range of SES indicators–all of which have downstream consequences for health [5, 11, 15]. These include income, employment opportunity, as well as access to quality education, housing, and health care [16–21]. To this point, Massey and Fischer [5] argued that racialized segregation interacts with structural shifts in population-wide SES factors (e.g., increasing income inequality or class segregation) to augment spatial concentrations of poverty in racial and ethnic minority populations. This suggests that racialized residential segregation represents a key determinant of population health disparities because it restricts socioeconomic capital and mobility in racial and ethnic minority populations by increasing wealth in majority-white populations [5, 22]. Yet, to date, the majority of research has fallen short of directly testing this pathway. Most empirical studies in this area have conceptualized segregation in unidimensional terms of either SES markers *or* the racial make-up of a population in a given area [10, 23], often overlooking the importance of the interaction of SES and race/ethnicity [8, 10, 11, 24–26]. These limitations are symptomatic of the wider evidence base on segregation and health, which generally contains little empirical attention to the pathways that connect the main variables of interest [10, 11, 25].

### Index of concentration at the extremes

In response to the limitations in the empirical conceptualization and operationalization of residential segregation, emerging research has begun to utilize a more comprehensive method for explaining the segregation-health relationship. Developed by Massey [22] in 2001, the Index of

Concentration at the Extremes (ICE) measures the extent to which the population of a given area is concentrated into relative extremes of advantage and deprivation. Formulaically, the measure is defined as $ICE_i = (A_i - P_i)/T_i$ where $A_i$ might represent the number of affluent persons in neighborhood $i$ (e.g., in the 80th income percentile), $P_i$ the number of poor persons in neighborhood $i$ (e.g., in the 20th income percentile), and $T_i$ the total population in neighborhood $i$ with known income. The ICE is thus scaled from -1 to +1 where -1 indicates that 100% of the population in the given area is concentrated in the most deprived group, and +1 means that 100% of the population is concentrated in the most privileged group.

In 2015, building on the work of Massey and Fischer [5, 25], Krieger, Waterman, Gryparis and Coull [27] took a novel approach to the measure and applied it to race, income, and their interaction. This innovation included three separate ICE measures. For $ICE_{Income}$, cut-off points were set at the 80th and 20th income percentiles for highest- (most privileged) and lowest-income (most deprived) areas, respectively. For $ICE_{Race}$, the areas with the highest concentrations of white residents represented the most privileged areas, while high spatial concentrations of racial and ethnic minorities (e.g., Black population) represented the most deprived areas. The last ICE, $ICE_{Race-Income}$, combined data on income and race, with the privileged extreme denoting spatial concentrations of white residents in the 80th income percentile vs. the deprived extreme of spatial concentrations of Black residents in the 20th income percentile. In this way, Krieger et al. adapted the original unidimensional ICE to a multidimensional metric ($ICE_{Race-Income}$) of *racialized economic segregation.*

The $ICE_{Race-Income}$ may have several advantages over other more conventional measures of segregation. First, by simultaneously accounting for spatial *and* social polarization, it is more comprehensive than other popular segregation measures which tend to focus on either spatial *or* social segregation [28]. In doing so, the $ICE_{Race-Income}$ also avoids common multi-collinearity issues associated with using separate measures of advantage and deprivation [22]. Another notable feature of the $ICE_{Race-Income}$ relates to its ready applicability to geographic areas of varying size (e.g., city block level or census tract to city or region). By contrast, the most commonly used measures of segregation (e.g., the Index of Dissimilarity and the Gini Coefficient for racial and economic segregation, respectively [10]) are not particularly informative at the neighborhood or census tract level of measurement–arguably the most meaningful spatial contexts in which to examine the health effects of segregation [10]. For example, neighborhoods that are entirely low- or high-income would have the same Gini coefficient given their perfect income equality. Similarly, neighborhoods that are entirely white or Black would have the same Index of Dissimilarity score on account of residents belonging to only one of two groups in focus. In other words, in contrast to the $ICE_{Race-Income}$, the utility and accuracy of these types of 'evenness measures' of segregation breaks down when applied to smaller geographical areas. Finally, the $ICE_{Race-Income}$ also has the advantage of computing concentration directionality (i.e., -1 to +1) rather than merely indicating whether unequal distributions are present or not.

## Rationale

This review focuses on the link between racialized economic segregation, as operationalized by $ICE_{Race-Income}$, and health disparities in the US. The evidence base on the association between segregation and health is extensive, but not exhaustive. As noted, a key limitation relates to the lack of knowledge on the mechanisms that underpin the observed relationship between segregation and health. Gaining insight into these pathways is crucial in terms of identifying and disrupting the processes by which segregation leads to deleterious health outcomes in racial and ethnic minority populations. Consistent with these knowledge gaps, Acavedo-Garcia,

Lochner, Osypuk and Subramanian [9] published a review of the literature in 2003. Here, the authors concluded that there was a distinct need for better ways of quantifying segregation effects on health. Specifically, they called for multidimensional pathway measures that incorporated SES as well as spatial polarization. Kramer and Hogue echoed this sentiment in their review from 2009 [10]. By incorporating race, ethnicity and income into a single measure that can be applied at multiple levels of geography, $ICE_{Race-Income}$ represents an important step towards this goal. While this measure has steadily gained momentum in the literature as an improved way of operationalizing racialized economic segregation, no review has been conducted to evaluate the evidence-base linking $ICE_{Race-Income}$ to health outcomes. Thus, our central goal is to critically assess the literature as it relates to the association between health and $ICE_{Race-Income}$. Expressly, by consolidating the evidence base in this area, we aim to (1) contribute to a clearer picture of the mechanics of how racial segregation impacts on health, and (2) stimulate further research into the structural pathways through which racial segregation may impact on health (e.g., education, incarceration rates).

## Methods

### Protocol

The present review was conducted according to the Preferred Reporting Items for Systematic Reviews and Meta-Analyses (PRISMA) guidelines (PROSPERO ID: CRD42021261944). Full details can be access at www.prisma-guidelines.org.

### Literature search and inclusion strategy

We executed a rigorous and comprehensive search of the following EBSCOhost databases: Academic Search premier, Allied and Complementary Medicine Database, CINAHL Plus with Full Text, Global Health, Web of Science, and SocINDEX. We also conducted separate searches on PubMed and Cochrane Library and manually examined the bibliographies of relevant articles for additional references.

As noted above, we focused on the $ICE_{Race-Income}$ as a metric for racialized economic segregation and restricted our search to include only studies that looked at physical and mental health outcomes and/or health-risk biomarkers. For the database search, the exact Boolean/phrase literature-search syntax was as follows: "Structural racism" OR "Systemic racism" OR "Institutional racism" AND "Index of concentration at the extremes" OR "ICE" AND "Health". We refined the search results by adding limiters to include only peer reviewed studies in the English language that were published between January 1, 2001, and December 31, 2021. We restricted our search to these dates as the ICE measure was introduced in 2001 [22]. On PubMed and Cochrane Library, our search terms were: "(Index of concentration at the extremes [Title/Abstract/Full-text])" with no limiters. We executed our initial search in May 2021. To avoid missing any subsequently published relevant articles, we re-ran the full search in August 2021. Once the search was executed, we retained articles based on the following inclusion criteria:

1. The article reported empirical studies on racialized economic segregation and health,

2. Racialized economic segregation was measured using the ICE and conceptualized in multidimensional terms (i.e., spatial and social polarization in terms of race or ethnicity *and* income),

3. Health was defined as physical or mental health and measured in terms of general health outcomes, specific disease incidence, and/or health-risk biomarkers,

4. The research focused on US populations,

5. The research reported peer-reviewed quantitative results,

6. The full text was available in English.

After deduplication, each unique database hit was evaluated by the first and second authors in three rounds against the inclusion criteria. In the first round, articles that obviously were not relevant were discarded (usually based on title). In the second round, we examined article abstracts. Again, the articles that clearly did not relate to our subject matter were rejected. Finally, the articles that remained after the first two evaluation rounds were downloaded and scrutinized in full-text detail for relevance. Only papers that passed through each of these three rounds were included in the review.

## Data extraction and quality appraisal

The research quality appraisal was conducted by the first and second authors and two research assistants. To this end, a coding protocol detailing the exact information to be gleaned from the articles was implemented. The protocol also included a commonly used research quality assessment tool, the Mixed Methods Appraisal Tool (MMAT). For quantitative non-randomized research, the MMAT evaluates studies along five dimensions, including population representativeness, appropriateness of measures of exposure and outcome, completeness of outcome data, the extent to which relevant confounders were accounted for, and whether the intervention exposure was administered as intended. Given the fact that the exposure of interest for this review (i.e., segregation) occurs at the population level as opposed to the individual level, the last MMAT item (intervention exposure) was omitted for lack of relevance. Each dimension was assessed in terms of whether a given criteria had been met and scored yes/no/can't tell as appropriate. For the purposes of this review, we gave each paper a single score of 'high' (all dimensions coded 'yes'), 'medium' (all but one dimension coded 'yes'), or 'low' quality (two or fewer dimensions coded 'yes'). To ensure accuracy and completeness, the coding and appraisal protocol was implemented twice for each article by two different researchers and subsequently compared. Any discrepancies were resolved through discussion and re-examination of the given paper.

## Results

### Literature search results

Our initial search of the literature returned 2,604 hits. With the application of all database limiters and following deduplication, 504 articles were identified as being of potential relevance (see Fig 1). Of these, the majority was rejected based on one or more of the following reasons: The paper did not implement the Krieger et al. version of the ICE; the paper focused on interpersonal rather than structural discrimination; the paper did not cite empirical research (e.g., editorial, comment); the paper lacked sufficient statistical detail; or a combination of these. A total of 20 papers was retained for the review.

### Study characteristics and methodology

All the papers included in the present review were published in the past six years (as of 2021). In terms of study populations, papers focused on either white vs. Black ($n = 20$) and/or white vs. Hispanic populations ($n = 2$). For those studies that reported population sample sizes ($n = 16$), the average sample size was $N = 597,794$. Geographically, the reviewed studies were mainly representative of the Northern and Western states with only three studies conducted in the South (D.C., Louisiana, and Florida). Studies were most frequently conducted in New York City ($n = 6$),

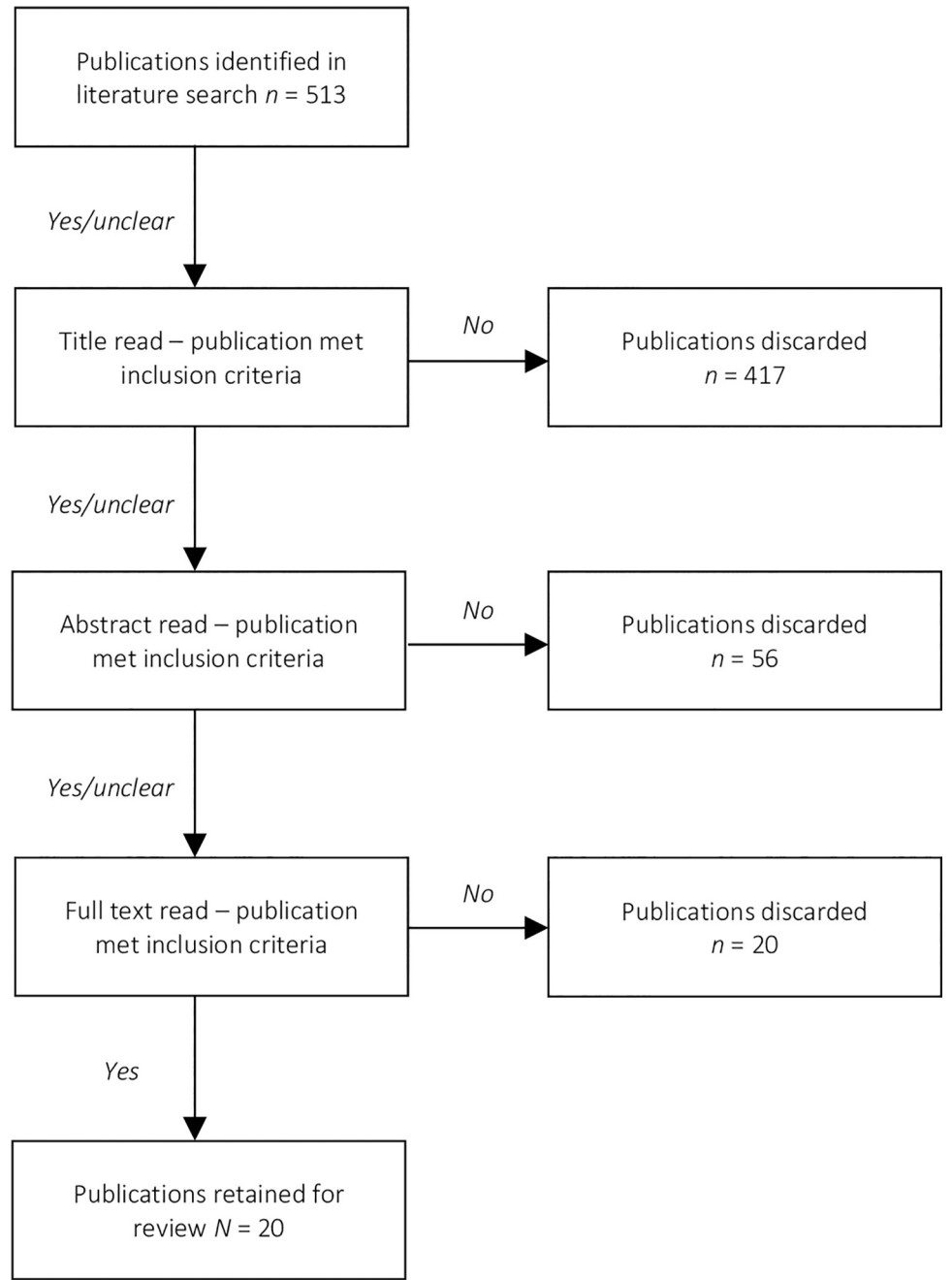

**Fig 1. Flow chart for each step of article evaluation and retainment.**

followed by locations in Massachusetts (*n* = 4), Illinois (*n* = 3), and California (*n* = 2). One study was based on a national sample [24]. Other study sites included New Jersey (*n* = 1), Louisiana (*n* = 1), Florida (*n* = 1), Michigan (*n* = 1), and Washington D.C. (*n* = 1). The area of measurement across studies was defined at multiple levels, including census tract (CT) (*n* = 11), followed by zip code (*n* = 4), county (*n* = 3), community district (CD) (*n* = 3), and city (*n* = 1). Four studies included measurements at multiple levels. Study designs were limited to two types: Cross-sectional population-based (*n* = 18) and cohort (*n* = 2) (see Table 1).

**Table 1. Study characteristics of the reviewed research.**

| Author (year) | Location | Population (N) | Research design | Data sources | Covariates | | Outcome | Research quality |
|---|---|---|---|---|---|---|---|---|
| **Bishop-Royse et al. (2021)** | Chicago, IL | Black/white adults (77 community districts), | Cross-sectional | Illinois Department of Health; Chicago Health Atlas; ACS | • Hardship scores,<br>• Household composition,<br>• Healthcare access. | | • Infant mortality rates | High |
| **Brown et al. (2021)** | Washington, D.C. | Black/white adults (705,000), | Cross-sectional | ACS; Government of District of Columbia's coronavirus website | N/A | | • Covid-19 incidence,<br>• Number of covid-19 tests,<br>• Covid-19 positivity rate | Medium |
| **Chambers et al. (2019)** | CA | Black/white singleton births (47,771) | Cross-sectional | California Birth Cohort files; ACS | • Age,<br>• Education,<br>• Nativity,<br>• Insurance,<br>• WIC use,<br>• Prenatal care visits,<br>• BMI,<br>• Smoking, | • Alcohol,<br>• Illicit drugs,<br>• Infection,<br>• Diabetes,<br>• Hypertension,<br>• Depression,<br>• Previous PTB | • Preterm birth,<br>• Infant mortality | High |
| **Chen & Krieger (2021)** | New York City, NY; IL | Black/white adults (68,656) | Cross-sectional | USA Facts; IL Dept. of Public Health; The Chicago Reporter; NYC Dept. of Health and Mental Hygiene; ACS | • Age,<br>• Sex,<br>• County characteristics | | • US county COVID-19 death rate,<br>• Zip code death rates IL,<br>• Zip code positivity rate NY. | Medium |
| **Dyer et al. (2021)** | LA | Black/white women who had given birth 2016–2017 (125,537) | Cross-sectional | Louisiana Dept. of Health; ACS | • Maternal age,<br>• Education,<br>• Access to resources,<br>• Urban vs. rural, | • Maternal race/ethnicity | • Pregnancy-related death | High |
| **Feldman et al. (2015)** | Boston, MA | Black/white adult union members (2,145) | Cross-sectional | United for Health; My Body My Story; ACS | • Race/ethnicity,<br>• Age,<br>• Gender,<br>• Smoking, | • BMI,<br>• Income,<br>• Education,<br>• Self-reported racism | • Hypertension | High |
| **Huynh et al. (2018)** | New York City, NY | Black/white singleton births (532,806) | Cross-sectional | New York City Dept. of Health and Mental Hygiene; ACS | • Maternal age,<br>• Infant sex,<br>• Maternal race/ethnicity,<br>• Maternal education, | • Marital status,<br>• Maternal insurance,<br>• WIC use,<br>• Maternal nativity,<br>• BMI | • Preterm birth<br>• Infant mortality | High |
| **Janevic et al. (2020)** | New York City, NY | Black/white women who gave birth at NYC hospital 2012–2014 (316,600) | Cross-sectional | Statewide Planning and Research Cooperative System; ACS | • Age,<br>• Education,<br>• Race/ethnicity,<br>• Nativity,<br>• Previous live births,<br>• BMI,<br>• Pre-pregnancy diabetes & hypertension,<br>• Gestational diabetes & hypertension,<br>• Cardiac disease,<br>• Renal disease, | • Pulmonary disease,<br>• Musculoskeletal disease,<br>• Blood disorders,<br>• Mental disorders,<br>• Central nervous system disorders,<br>• Rheumatic heart disease,<br>• Placental disorders,<br>• Anemia,<br>• Asthma,<br>• Prior cesarean delivery<br>• Hospital of delivery | • Severe Maternal Morbidity | High |

*(Continued)*

**Table 1.** (*Continued*)

| Author (year) | Location | Population (N) | Research design | Data sources | Covariates | | Outcome | Research quality |
|---|---|---|---|---|---|---|---|---|
| **Janevic et al. (2021)** | New York City, NY | Black/white infants born <32 weeks in 2010–2014 (6,461) | Cross-sectional | Statewide Planning and Research Cooperative System; ACS | • Maternal age, • Mother's education, • Smoking, • Insurance, • Multiparous, • Cesarean, • Infant sex, • Chorioamnionitis, | • Precipitous labor, • Placental abruption, • Pre-pregnancy hypertension, • Gestational hypertension, • Pre-pregnancy diabetes, • Gestational diabetes | • Morbidity and mortality in preterm neonates | High |
| **Krieger, Singh, et al. (2016)** | U.S. | Black/white women with primary invasive breast cancer (516,382) | Cross-sectional | US National Cancer Institute Surveillance, Epidemiology, and End Results (SEER) cancer registry; ACS | • Year of diagnosis, • Age at diagnosis, • Race/ethnicity, • Tumor size, | • Stage at diagnosis, • Histologic type, • Grade | • Breast cancer estrogen receptor (ER) status | High |
| **Krieger, Waterman, et al. (2016)** | New York City, NY | Black/white adults (59 CDs, 2,126 CTs) | Cross-sectional | ACS; New York City Dept. of Health and Mental Hygiene | N/A | | • Infant mortality, • Diabetes mortality, • All-cause mortality | Medium |
| **Krieger et al. (2017)** | Boston, MA | Black/white adults (15 neighborhoods, 170 census tracts) | Cross-sectional | Geocoded birth and death data from Massachusetts Dept. of Public Health; ACS | N/A | | • Preterm birth, • Premature mortality (<65yrs) | Medium |
| **Krieger, Feldman, et al. (2018)** | MA | Black/white adults (6,540,189) | Cross-sectional | ACS; Massachusetts Cancer Registry | • Age, • Sex, • Urbanicity, • City/town characteristics, | | • Black/white cancer incidence | High |
| **Krieger, Kim, et al. (2018)** | MA | Black/white decedents (263,266) | Cross-sectional | Massachusetts Dept. of Public Health; ACS | • Gender, • Race/ethnicity, • Urbanicity | | • Mortality outcomes (child <5yrs; premature <65yrs; cause-specific) | High |
| **Krieger et al. (2020)** | New York City, NY | Black/white singleton births 2013–2017 (528,096) | Cross-sectional | NYC Dept. of Health and Mental Hygiene vital statistics birth certificate data; 1938 HOLC grade; ACS | • Maternal race/ethnicity, • Age at giving birth, • Nativity, • Education | | • Preterm birth | High |
| **Lange-Maia et al. (2018)** | Chicago, IL | Black/white adults (77 Chicago Community Areas) | Cross-sectional | Chicago Dept. of Public Health; ACS | N/A | | • Premature mortality | Medium |
| **Shrimali et al. (2020)** | CA | Black/white singleton births (379,794) | Cohort | California Biobank Program's biobank linked database; ACS | • Maternal age, • Race/ethnicity, • Education, • Public insurance, | • Mother's poor birth outcome at own birth. | • Preterm delivery | High |
| **Wallace et al. (2019)** | Wayne County, MI | Black/white births between 2010 and 2013 (84,159) | Cross-sectional | The Michigan Dept. of Health and Human Services Vital Records Division; ACS 2009–2013 | • Maternal age, • Marital status, • Plurality, • Insurance type | | • Infant mortality | High |
| **Westrick et al. (2020)** | FL | Black/white & Hispanic vs. white women diagnosed with EOC (16,431) | Cross-sectional | Florida Cancer Data System; ACS 2012–2016 | • Age, • Insurance, • Histology, • Tumor stage, | • Surgery, • Chemo, • Deceased | • EOC survival rate | High |

(*Continued*)

**Table 1.** (Continued)

| Author (year) | Location | Population (N) | Research design | Data sources | Covariates | | Outcome | Research quality |
|---|---|---|---|---|---|---|---|---|
| **Wiese et al. (2019)** | NJ | Black/white/ Hispanic women with breast cancer (27,078) | Cohort | New Jersey State Cancer Registry; ACS 2011–2014 | • Age, • Vital status, • Stage at diagnosis, • BC subtype, | • Marital status, • Insurance | • Breast cancer survival | High |

In keeping with the focus of the review, all the included studies employed the $ICE_{Race-income}$ as a measure of racialized economic segregation, the main predictor variable. Most studies also included single-component measures of racial ($ICE_{Race}$, $n = 17$) and/or economic ($ICE_{Income}$, $n = 18$) residential segregation. The ICE measures were operationalized in an identical fashion across studies to measure:

1. Racial/ethnic residential segregation ($ICE_{Race}$) in terms of relative spatial concentrations of privileged (white) vs. disadvantaged (racial/ethnic minority) populations,

2. Economic residential segregation ($ICE_{Income}$) in terms of relative spatial concentrations of high-income (in the $80^{th}$ income percentile) vs. low-income (in the $20^{th}$ income percentile) populations, and

3. Racialized economic segregation ($ICE_{Race-Income}$) in terms of relative spatial concentrations of privileged high-income (white residents in the $80^{th}$ income percentile) vs. disadvantaged low-income (racial/ethnic minorities in the $20^{th}$ income percentile) populations.

ICE measures were calculated based on American Community Survey (ACS) data and arranged in quantiles (typically quartiles or quintiles) with the low extreme (-1) representing the most disadvantaged and the high extreme (+1) the most privileged. Unless otherwise specified, the following results describe the relationship between the two extreme quantiles (e.g., area with the highest vs. lowest average income, highest concentration of Black residents vs. highest concentration of white residents).

In terms of outcomes, studies focused on adverse birth outcomes (preterm birth, infant mortality, maternal death, $n = 11$), cancer outcomes ($n = 4$), premature mortality ($n = 4$), COVID-19 outcomes ($n = 2$), and/or hypertension ($n = 1$).

## Research quality

The inter-rater MMAT research quality assessment was well-aligned among the four coders with conflicts on only five papers (inter-rater reliability = 77.3%). Each of these discrepancies was resolved by revisiting the article in question and discussing the point of divergence. Fifteen (75%) papers were appraised as being of 'high' quality and five (25%) were 'medium' quality. As such, most articles fulfilled all MMAT criteria, implementing rigorous research designs and leveraging high-quality secondary data that reflected representative study populations pertinent to the study focus. Medium ratings were typically due to lacking information about covariates.

## Study findings

In the following sections we briefly describe the main results from each of the papers. Detailed statistics for the primary variables for each study are provided in Table 2. Unless otherwise stated, the results summarized here are adjusted for all relevant covariates included in each

**Table 2. ICE and poverty measure statistics.**

| Author | Outcome | Racial/ethnic contrast | Geo level | ICE$_{Race}$ | ICE$_{Income}$ | ICE$_{Race-Income}$ | Poverty/HI measure |
|---|---|---|---|---|---|---|---|
| *Adverse birth/pregnancy outcomes* | | | | | | | |
| Bishop-Royse et al. (2021) | Infant mortality | Black/white | CD | IRR = 0.46** | IRR = 0.23** | IRR = 0.21** | – |
| Chambers et al. (2019) | PTB | Black/white | Zip code | OR = 1.15, CI 1.02, 1.30* | OR = 1.29, CI 1.16, 1.44* | OR = 1.25, CI 1.12, 1.40* | – |
| | Infant mortality | | | OR = 1.54, CI 1.03, 2.30* | OR = 1.41, CI 0.91, 2.48 | OR = 1.68, CI 1.14, 2.47* | |
| Dyer et al. (2021) | Maternal death | Black/white | CT | – | – | RR = 1.17, CI 0.62, 2.19 | – |
| Huynh et al. (2018) | PTB | Black/white | CT | OR = 1.41, CI 1.34, 1.49+ | OR = 1.16, CI 1.10, 1.21+ | OR = 1.36, CI 1.29, 1.43+ | OR = 1.09, CI 1.04, 1.14+ |
| | Infant mortality | | | OR = 1.80, CI 1.43, 2.28+ | OR = 1.18, CI 0.97, 1.43+ | OR = 1.54, CI 1.23, 1.94+ | OR = 1.09, CI 0.90, 1.32+ |
| Janevic et al. (2020) | SMM | Black/white | Zip code | RD = 2.40, CI 2.00, 2.80+ | RD = 1.40, CI 0.80, 2.00+ | RD = 2.30, CI 1.90, 2.70+ | – |
| Janevic et al. (2021) | Neonatal mortality/ morbidity | Black/white | Neighborhood | OR = 1.60, CI 1.20, 2.10+ | OR = 1.40, CI 1.10, 1.90+ | OR = 1.59, CI 1.20, 2.20+ | – |
| Krieger et al. (2020) | PTB | Black/white | CT | – | – | RR = 1.25, CI 1.20, 1.30* | – |
| Krieger, Waterman, et al. (2016) | Infant mortality | Black/white | CT | RR = 2.77, CI 2.02, 3.81+ | RR = 2.19, CI 1.59, 3.02+ | RR = 2.93, CI 2.11, 4.09+ | RR = 1.56, CI 1.19, 2.04+ |
| | | | CD | RR = 2.19, CI 1.89, 2.53+ | RR = 2.66, CI 2.33, 3.05+ | RR = 2.57, CI 2.21, 2.99+ | RR = 1.99, CI 1.70, 2.32+ |
| Krieger et al. (2017) | PTB | Black/white | CT | OR = 1.20, CI 1.09, 1.33* | OR = 1.14, CI 1.03, 1.26* | OR = 1.19, CI 1.08, 1.31* | RR = 1.10, CI 0.99, 1.22* |
| | | | Neighborhood | OR = 1.26, CI 1.14, 1.39* | OR = 1.09, CI 0.98, 1.20* | OR = 1.17, CI 1.06, 1.29* | RR = 1.07, CI 0.97, 1.18* |
| Shrimali et al. (2020) | PBT | Black/white | CT | RR = 1.02, CI 0.98, 1.06$^{CH+}$ | RR = 1.10, CI 1.06, 1.14$^{CH+}$ | RR = 1.12, CI 1.08, 1.17$^{CH+}$ | – |
| | | | | RR = 1.04, CI 1.00, 1.08$^{AH+}$ | RR = 1.11, CI 1.07, 1.15$^{AH+}$ | RR = 1.07, CI 1.03, 1.11$^{AH+}$ | |
| Wallace et al. (2019) | Infant mortality | | CT | – | – | OR = 1.46, CI 1.02, 2.09* | – |
| *Cancer outcomes* | | | | | | | |
| Krieger, Feldman, et al. (2018) | Cervical cancer | Black/white | CT | IRR = 2.54, CI 1.75, 3.68+ | IRR = 2.61, CI 1.85, 3.67+ | IRR = 3.02, CI 2.13, 4.27+ | IRR = 1.88, CI 1.38, 2.55+ |
| | | | City/town | IRR = 0.84, CI 0.55, 1.29+ | IRR = 1.19, CI 0.81, 1.73+ | IRR = 0.96, CI 0.67, 1.38+ | IRR = 1.29, CI 0.92, 1.82+ |
| | Lung cancer | | CT | IRR = 1.44, CI 1.31, 1.59+ | IRR = 1.48, CI 1.36, 1.61+ | IRR = 1.52, CI 1.40, 1.66+ | IRR = 1.49, CI 1.39, 1.60+ |
| | | | City/town | IRR = 1.12, CI 0.99, 1.28+ | IRR = 1.39, CI 1.25, 1.55+ | IRR = 1.40, CI 1.26, 1.55+ | IRR = 1.45, CI 1.31, 1.60+ |
| | Breast cancer | | CT | IRR = 1.01, CI 0.95, 1.08+ | IRR = 0.86, CI 0.82, 0.91+ | IRR = 0.89, CI 0.84, 0.94+ | IRR = 0.88, CI 0.83, 0.93+ |
| | | | City/town | IRR = 1.09, CI 1.02, 1.16+ | IRR = 0.86, CI 0.82, 0.90+ | IRR = 0.86, CI 0.81, 0.90+ | IRR = 0.90, CI 0.85, 0.95+ |
| Krieger, Singh, et al. (2016) | ER status | Black/white | County | OR = 1.27, CI 1.11, 1.45+ | OR = 1.14, CI 1.05, 1.24+ | OR = 1.24, CI 1.07, 1.43+ | – |
| Westrick et al. (2020) | Ovarian cancer mortality | Black/white | Neighborhood | HR = 1.12, CI 1.02, 1.22* | HR = 1.15, CI 1.06, 1.25* | HR = 1.21, CI 1.12, 1.32* | – |
| | | Hispanic/white | | HR = 1.02, CI 0.93, 1.11* | – | HR = 1.12, CI 1.03, 1.22* | |
| Wiese et al. (2019) | Breast cancer death | Black/white | Geo clusters | Results reported in text | Results reported in text | Results reported in text | – |
| | | Hispanic/white | | | | | |
| *Premature mortality* | | | | | | | |

(Continued)

**Table 2.** (Continued)

| Author | Outcome | Racial/ethnic contrast | Geo level | ICE$_{Race}$ | ICE$_{Income}$ | ICE$_{Race-Income}$ | Poverty/HI measure |
|---|---|---|---|---|---|---|---|
| Lange-Maia et al. (2018) | Premature mortality | Black/white | CD | RR = 3.07, CI 2.62, 3.58[+] | RR = 3.06, CI 2.51, 3.73[+] | RR = 3.27, CI 2.84, 3.77[+] | RR = 2.79, CI 2.18, 3.57[+] |
| Krieger, Waterman, et al. (2016) | Premature mortality | Black/white | CT | RR = 1.89, CI 1.79, 2.00[+] | RR = 2.24, CI 2.12, 2.37[+] | RR = 2.33, CI 2.21, 2.46[+] | RR = 2.10, CI 2.00, 2.20[+] |
| | | | CD | RR = 1.78, CI 1.74, 1.82[+] | RR = 2.36, CI 2.30, 2.42[+] | RR = 2.26, CI 2.20, 2.32[+] | RR = 2.40, CI 2.33, 2.47[+] |
| | Diabetes mortality | | CT | RR = 2.78, CI 2.37, 3.26[+] | RR = 2.85, CI 2.43, 3.36[+] | RR = 3.52, CI 3.00, 4.12[+] | RR = 2.76, CI 2.39, 3.19[+] |
| | | | CD | RR = 2.96, CI 2.75, 3.19[+] | RR = 3.17, CI 2.92, 3.45[+] | RR = 3.79, CI 3.50, 4.11[+] | RR = 3.49, CI 3.20, 3.80[+] |
| Krieger et al. (2017) | Premature mortality | Black/white | CT | RR = 1.66, CI 1.43, 1.93** | RR = 1.58, CI 1.36, 1.83** | RR = 1.63, CI 1.40, 1.90** | RR = 1.47, CI 1.27, 1.71** |
| | | | Neighborhood | RR = 1.42, CI 1.23, 1.63** | RR = 1.46, CI 1.03, 2.09** | RR = 1.39, CI 1.19, 1.61** | RR = 1.33, CI 1.15, 1.54** |
| Krieger, Kim, et al. (2018) | Child disease mortality | Black/white | CT | RR = 1.85, CI 1.33, 2.57[+] | RR = 1.64, CI 1.20, 2.23[+] | RR = 2.19, CI 1.60, 3.00[+] | RR = 1.40, CI 1.07, 1.83[+] |
| | | | City/town | RR = 1.13, CI 0.76, 1.68[+] | RR = 1.40, CI 1.01, 1.93[+] | RR = 1.13, CI 0.81, 1.57[+] | RR = 1.61, CI 1.19, 2.16[+] |
| | Adult disease mortality | | CT | RR = 2.28, CI 2.06, 2.52[+] | RR = 2.30, CI 2.13, 2.49[+] | RR = 2.39, CI 2.21, 2.59[+] | RR = 2.01, CI 1.86, 2.17[+] |
| | | | City/town | RR = 0.97, CI 0.84, 1.13[+] | RR = 1.51, CI 1.36, 1.67[+] | RR = 1.53, CI 1.39, 1.69[+] | RR = 1.38, CI 1.23, 1.55[+] |
| *Other health outcomes* | | | | | | | |
| Brown et al. (2021) | SARS-CoV-2 incidence[1] | Black/white | Neighborhood | $\rho$ = -0.59*** | $\rho$ = -0.46*** | $\rho$ = -0.53*** | – |
| | SARS-CoV-2 positive tests[1] | | | $\rho$ = -0.81*** | $\rho$ = -0.64*** | $\rho$ = -0.72*** | |
| | SARS-CoV-2 testing rates[1] | | | $\rho$ = 0.30* | $\rho$ = 0.33* | $\rho$ = 0.32* | |
| | SARS-CoV-2 incidence[2] | | | $\rho$ = -0.53*** | $\rho$ = -0.56*** | $\rho$ = -0.61*** | |
| | SARS-CoV-2 positive tests[2] | | | $\rho$ = -0.80*** | $\rho$ = -0.77*** | $\rho$ = -0.84*** | |
| | SARS-CoV-2 testing rates[2] | | | $\rho$ = 0.54* | $\rho$ = 0.38** | $\rho$ = 0.45** | |
| Chen & Krieger (2021) | SARS-CoV-2 death rate | Black/white | County | – | – | RR = 1.04, CI 1.02, 1.06[+] | |
| | SARS-CoV-2 cases | | ZCTA | – | – | RR = 3.19, CI 3.19, 3.27[+] | – |
| | SARS-CoV-2 positive tests | | ZCTA | – | – | RR = 1.68, CI 1.65, 1.71[+] | |
| Feldman et al. (2015) | Hypertension | Black/white | CT | OR = 0.76, CI 0.62, 0.93* | – | OR = 0.48, CI 0.29, 0.81* | – |
| | | POC/white | | – | – | OR = 0.61, CI 0.40, 0.96* | |

Note.

* = < .05

** = < .01

*** = < .001

+ = Exact p-value not reported. All CIs = 95%; RR = Risk ratio, RD = Risk difference, IRR = Incidence risk ratio, HR = Hazard ratio, OR = Odds ratio, CH = Childhood, AH = Adulthood

[1] First six months of 2020

[2] Last six months of 2020.

study (see Table 1). Further, unless specified, 'Black' and 'white' populations are of non-Hispanic ethnicity. Given the clustering of study focus among the articles, we divide this section into the following four sub-sections defined by outcome: (1) adverse birth and pregnancy outcomes, (2) cancer outcomes, (3) premature mortality, and (4) other health outcomes (COVID-19, hypertension). Some articles focus on several outcomes (e.g., preterm birth and premature mortality) and thus feature in multiple subsections.

## Racialized economic segregation and adverse birth/pregnancy outcomes

Eight articles examined adverse birth and pregnancy outcomes. Preterm birth (PTB; < 37 weeks gestational age) and infant mortality (death of child under the age of one year) were the most frequent outcomes of interest, with one or the other, or both measured in all relevant studies. Other outcomes included infant morbidity ($n$ = 1) and/or maternal morbidity or death due to pregnancy complications ($n$ = 2).

Shrimali, Pearl, Karasek, Reid, Abrams and Mujahid [29] investigated whether exposure to racialized economic segregation in early childhood and adulthood were independently associated with racial disparities in PTB-risk. Controlling for either childhood or adulthood exposure, the authors found that women who spent their childhood in low-income census tracts (CT) had a 10% increased risk of PTB compared to women who had spent their childhood in high-income CTs (childhood $ICE_{Income}$). There was no statistical difference in PTB risk for childhood exposure to racialized segregation (childhood $ICE_{Race}$). Childhood exposure to low-income majority-Black CTs as opposed to high-income majority-white CTs (childhood $ICE_{Race-Income}$), however, was associated with a 12% increased risk of PTB. In terms of adult exposure, low-income as opposed to high-income CTs was associated with an 11% higher risk of PTB (adult $ICE_{Income}$). Similarly, there was a 4% increased risk of PTB associated with adult exposure to majority-Black as opposed to majority-white CTs (adult $ICE_{Race}$). Finally, adult exposure to low-income majority-Black CTs compared to high-income white CTs was associated with a 7% increased risk of PTB (adult $ICE_{Race-Income}$). The findings indicate independent positive associations between PTB risk and both childhood and adulthood exposures to racialized economic segregation. This suggests immediate as well as long-term harms of this type of segregation.

In a similar study on redlining and PTB in New York City, Krieger, Van Wye, Huynh, Waterman, Maduro et al. [30] found that women living in historically redlined vs. 'green' CTs (the most desirable grade designated by the Home Owners' Loan Corporation [HOLC]) were 55% more likely to experience PTB (OR = 1.55, 95% CI 1.39, 1.72). However, this association attenuated to a non-significant level when controlling for current spatial concentrations of high-income white vs. low-income Black residents. Specifically, in low-income majority-Black CTs, PTB rates were 25% higher than in high-income majority-white CTs ($ICE_{Race-Income}$). While the authors did not assess mediation directly, this suggests that the relationship between past HOLC grade and PTB, may be the consequential effects of historical redlining policies that persist to the present day and continue to impact negatively on racial minority populations.

These results align with an earlier study by Krieger, Waterman, Batra, Murphy, Dooley, et al. [31] conducted in 15 city neighborhoods and 170 CTs in Boston, MA. Here, the authors found that residents in low- as opposed to high-income neighborhoods were 9% more likely to give birth preterm ($ICE_{Income}$). At the CT level, the odds increased to 14% ($ICE_{Income}$). Similarly, in majority-Black neighborhoods and CTs ($ICE_{Race}$), the odds of PTB were 26% and 20% higher than in majority-white neighborhoods and CTs, respectively. Combining racial and economic segregation ($ICE_{Race-Income}$), residents in low-income majority-Black neighborhoods

and CTs had 17% and 19% higher odds, respectively, of PTB than high-income majority-white neighborhoods and CTs. Importantly, the authors also included a poverty measure, derived from the ACS, as a comparison predictor variable. All three ICE measures at both levels of geography outperformed this measure based on magnitude of effect size.

Another study by Chambers, Baer, McLemore, and Jelliffe-Pawlowski [32] investigated the extent to which racialized economic segregation was associated with increased risk of PTB and infant mortality in white and Black populations in California. Operationalizing segregation at the zip-code level, the authors found that women who lived in the poorest areas were 29% more likely than women in the most affluent areas to experience PTB (ICE$_{Income}$). Similarly, women in majority-Black zip codes (ICE$_{Race}$) were 15% more likely than their counterparts in majority-white zip codes to give birth preterm. Finally, for women in low-income majority-Black areas, the odds of PTB were 25% higher than for women in high-income majority-white areas (ICE$_{Race-Income}$). The odds for infant mortality were comparable. Women in majority-Black areas were 54% more likely than women in white areas to experience infant death (ICE$_{Race}$), while women in low-income majority-Black areas were 68% more likely than women in high-income majority-white areas to experience infant death (ICE$_{Race-Income}$). No independent association of income with infant mortality was detected.

Next, Huyhn, Spasojevic, Li, Maduro, Van Wye, Waterman, and Krieger [33] conducted a study at the CT-level in New York City and found that the odds of PTB were 16% higher for populations in low-income vs. high-income CTs (ICE$_{Income}$), 41% higher in majority-Black vs. white CTs (ICE$_{Race}$), and 36% higher in low-income majority-Black vs. high-income majority-white CTs (ICE$_{Race-Income}$). Further, racial segregation was associated with an 80% increase in the odds of infant mortality (ICE$_{Race}$) while racialized economic segregation (ICE$_{Race-Income}$) predicted a 54% increase in odds of infant mortality. This study also included an ACS poverty measure. Unadjusted results indicated that the positive association between PTB and neighborhood poverty was significant, but considerably weaker than the ICE measures. In fully adjusted models, poverty was rendered non-significant. Similar to Chambers et al. [32], this study found no independent association between economic segregation (ICE$_{Income}$) and infant mortality.

In another paper on PTB and infant mortality, Janevic et al. [34] found that infants born to women in majority-Black neighborhoods had 60% greater odds of neonatal mortality and/or morbidity compared to women in majority-white neighborhoods (ICE$_{Race}$). Further, the odds of neonatal mortality and/or morbidity were 40% higher in low-income compared to high-income neighborhoods (ICE$_{Income}$) and 59% higher in low-income majority-Black neighborhoods compared to high-income majority-white neighborhoods (ICE$_{Race-Income}$). Importantly, the association between neonatal mortality/morbidity and ICE$_{Race}$ (though not ICE$_{Income}$ or ICE$_{Race-Income}$) was partially explained by the location of hospital of delivery. Women who lived in and gave birth at hospitals in majority-Black areas had an adjusted risk of neonatal morbidity and/or mortality of 38%. By contrast, the corresponding risk for women who lived in majority-Black neighborhoods but gave birth at hospitals in majority-white neighborhoods was 25%. This difference in risk was statistically significant, indicating that location of hospital of delivery partially mediated the observed association for ICE$_{Race}$ and neonatal morbidity/ mortality. In terms of maternal health, results also indicated a six to seven-fold increased risk of pre-pregnancy hypertension, a five to seven-fold increased risk of gestational hypertension, and a six to nine-fold increased risk of gestational diabetes for women in majority-Black, low-income, and low-income majority-Black neighborhoods. The magnitude of this association was slightly stronger for ICE$_{Race-Income}$. On this basis, the authors speculated that maternal comorbidity might also mediate the association between racialized economic segregation and birth outcomes.

Complementing the results in Janevic et al., Bishop-Royse, Lange-Maia, Murray, Shah, and DeMaio [35] conducted a study on infant mortality in 77 community areas in Chicago, IL. Here, majority-white communities had a 54% lower risk of infant mortality than majority-Black communities (ICE$_{Race}$). Similarly, the risk of infant mortality was 77% lower in high-income communities compared to low-income communities (ICE$_{Income}$) and 79% lower in high-income majority-white communities compared to low-income majority-Black communities (ICE$_{Race-Income}$). Adjusting for socioeconomic hardship (Hardship Index), proportion of single-parent households, and adequacy of prenatal care, ICE$_{Race}$ accounted for 46% of the variance in infant mortality rates followed by ICE$_{Race-Income}$ which accounted for 22% of the variance.

In a similar study on infant mortality rates in New York City CTs and community districts (CDs), Krieger, Waterman, Spasojevic, Li, Maduro, et al. [28] found that the risk of infant mortality was 119% greater in low-income CTs compared to high-income CTs (ICE$_{Income}$). In majority-Black CTs, infant mortality risk was 177% higher than in majority-white CTs (ICE$_{Race}$), and in low-income, majority-Black CTs the risk was 193% higher than in high-income majority-white CTs (ICE$_{Race-Income}$). At the CD-level, effect sizes were comparable, though slightly weaker (see Table 2). The authors also included a poverty measure (derived from the ACS) as a comparison predictor variable. At both geographical levels, the ICE measures accounted for more of the variance than this measure (see Table 2).

Further bolstering the findings on the general relationship between racialized economic segregation and infant mortality, Wallace, Crear-Perry, Green, Felker-Kantor, and Theall [36] conducted a cross-sectional study on infant mortality rates in Wayne County, MI. Results indicated disparities along racial and economic lines consistent with the evidence base. Particularly, the authors found that the odds of infant mortality in majority-Black, low-income CTs were 46% greater than in majority-white, high-income CTs (ICE$_{Race-Income}$). This study did not report independent ICE$_{Race}$ or ICE$_{Income}$ statistics.

The last two studies on birth- and pregnancy outcomes focused on pregnancy-related maternal death (up to one year post-partum) and morbidity. Dyer, Chambers, Crear-Perry, Theall, and Wallace [37] found that compared to women in high-income majority-white CTs, women in low-income majority-Black CTs had a 73% increased risk of maternal death after controlling for all covariates except maternal race (ICE$_{Race-Income}$). Controlling for maternal race attenuated the effect size to a 17% increased risk (ICE$_{Race-Income}$), indicating that the harmful effects of racialized economic segregation extend to all races within the deprived area. Similar to Janevic et al. [38], the authors also found that pre-pregnancy chronic comorbidities (diabetes and/or hypertension) mediated the observed association. Specifically, women in low-income majority-Black CTs were more than twice as likely as residents in high-income majority-white CTs to die from pregnancy-related issues that resulted from their chronic conditions (RR$_{IndirectEffect}$, RR = 2.68, 95% CI 2.59, 2.84). This was only a partial mediation, however. Keeping chronic disease constant across ICE terciles, women in low-income majority-Black CTs were still 51% more likely to die from pregnancy-related factors than their counterparts in high-income majority-white CTs (RR$_{IndirectEffect}$, RR = 1.51, 95% CI 1.18, 2.74).

Finally, Janevic, Zeitlin, Egorova, Hebert, Balbierz, et al. [39] examined the link between racialized economic segregation and severe maternal morbidity (SMM; having a life-threatening condition or life-saving procedure during childbirth). Their study was conducted across 183 zip codes in New York City. In unadjusted models, they found a 140% increased risk of SMM in majority-Black compared to majority-white zip codes (ICE$_{Race}$). Similarly, there was a 40% increased SMM risk in low-income compared to high-income zip codes (ICE$_{Income}$), and 130% increased risk in low-income majority-Black zip codes compared to high-income majority-white zip codes (ICE$_{Race-Income}$). Adjusting for all covariates, these differences attenuated

considerably ($ICE_{Race}$, RD = 0.3, 95% CI 0.1, 0.6; $ICE_{Income}$, RD = 0.4, 95% CI 0.2, 0.6; $ICE_{Race-Income}$, RD = 0.4, 95% CI 0.2, 0.6). Dovetailing with Janevic et al. [38], decomposition analyses indicated that the location of hospital of delivery accounted for 34.8% of the association between $ICE_{Race-Income}$ and SMM, with women in privileged or deprived areas more likely than not to deliver at local hospitals. Further, the presence of comorbidities accounted for nearly 50% of the variance in SMM. Finally, at the individual level, the authors also found that race/ethnicity moderated the link between $ICE_{Race-Income}$ and SMM, with risk differences being largest for Latina (RD = 1.6, 95% CI 1.0, 2.1), Black (RD = 1.6, 95% CI 0.4, 2.9), and Black-Latina women (RD = 2.6, 95% CI 1.2, 4.0).

## Racialized economic segregation and cancer outcomes

Four papers focused on cancer outcomes. In a study on breast, cervical, and lung cancer incidence at the CT and city/town level in Massachusetts, Krieger et al. [40] found a 161% increased risk of cervical cancer for residents in low-income vs. high-income CTs ($ICE_{Income}$) and a 154% increased risk for people in majority-Black as opposed to majority-white CTs ($ICE_{Race}$). Residents in low-income majority-Black CTs had a 202% increased risk of cervical cancer compared to people in high-income majority-white CTs ($ICE_{Race-Income}$). The risk pattern for lung cancer was similar. Residents in low-income vs. high-income and majority-Black vs. majority-white CTs had 48% ($ICE_{Income}$) and 44% ($ICE_{Race}$) increased risk of lung cancer, respectively. Residents in low-income majority-Black CTs had a 52% higher risk of lung cancer than their counterparts in high-income majority-white CTs ($ICE_{Race-Income}$). The pattern was similar at the city/town-level, though associations were statistically weaker. In terms of breast cancer, there was a 9% increased risk for residents in majority-Black as opposed to majority-white city/towns ($ICE_{Race}$). No other statistically significant associations between any of the ICE measures and breast cancer were evident at either geographical level (see Table 2). The authors also included a poverty measure based on the federal poverty line. At the CT level, the ICE measures generally accounted for more or comparable portions of the variance in each outcome as the poverty measure. However, the opposite was true at the city/town level (see Table 2).

Another paper by Krieger et al. [24] focused on racialized economic segregation and estrogen receptor (ER) status in breast cancer patients. ER represents an important biomarker for breast cancer outcomes with ER+ signifying a more curable tumor than ER-. Controlling for age and tumor characteristics, the authors found that patients who lived in high-income as opposed to low-income counties were 14% ($ICE_{Income}$) more likely to have an ER+ tumor, while women in majority-white vs. majority-Black counties were 27% ($ICE_{Race}$) more likely to have an ER+ tumor. Women in high-income majority-white counties were 24% more likely to have ER+ tumors than women in low-income, majority-Black counties ($ICE_{Race-Income}$). This is the only study included in this review that directly assessed the link between $ICE_{Race-Income}$ and variation in cellular-level health-risk factors. While the authors did not test for any mediators of the observed association, they speculated that the increased likelihood of ER- tumors in deprived communities were due to social ecological (i.e., environmental) factors.

In a similar study on epithelial ovarian cancer (EOC) survival ($N$ = 16,431), Westrick et al. [41] found that EOC-diagnosed women in low-income neighborhoods had a 15% increased risk of death compared to women in high-income neighborhoods ($ICE_{Income}$). For women in majority-Black as opposed to white neighborhoods, there was a 12% increased risk of death ($ICE_{Race}$). There was no statistical difference in risk of death from EOC for women in majority-Hispanic neighborhoods vs. women in majority-white neighborhoods. However, women living in low-income majority-Black neighborhoods as opposed to high-income majority-

white neighborhoods had a 21% higher risk of death from EOC (ICE$_{Race-Income}$). Women in low-income majority-Hispanic neighborhoods had a 12% higher risk of death than women in high-income majority-white neighborhoods (ICE$_{Race-Income}$).

Finally, Wiese et al. [42] investigated the link between racialized economic segregation and breast cancer survival rates in New Jersey. Based on Bayesian spatial models, the authors identified geographic clusters of death rates from breast cancer. In those clusters that had a statistically significant higher risk of death from breast cancer, 42.5% of the population lived in low-income areas and 11.5% in high-income areas (ICE$_{Income}$). Further, 50.2% lived in low-income majority-Black areas while only 7.1% lived in high-income majority-white areas (ICE$_{Race-Income}$). Nearly identical statistics were evident for low-income Hispanic neighborhoods vs. high-income white neighborhoods (50.2% vs. 6.8%, respectively). The authors also included a poverty measure based on the federal poverty line. The ICE measures explained slightly more (< 4%) of the geographic disparity in breast cancer survival than the poverty measure.

## Racialized economic segregation and premature mortality

Five studies focused on racialized economic segregation and premature mortality rates (death before age 65). Lange-Maia et al. [43] conducted a study in the 77 community areas of Chicago. Results indicated that residents in low-income communities had a 206% higher risk of premature mortality than residents in high-income communities (ICE$_{Income}$). In areas with the highest concentrations of Black as opposed to white population, there was an increased risk of 207% (ICE$_{Race}$). Low-income majority-Black community areas had a 227% increased risk of premature mortality compared to high-income majority white areas (ICE$_{Race-Income}$). The study also included the Hardship Index (HI). All three ICE measures each accounted for more of the variance than the HI.

Comparable results were generated by Krieger et al. [31]. The authors discovered a 46% and 58% increased risk of premature mortality in low- compared to high-income neighborhoods and CTs (ICE$_{Income}$), respectively. Comparable levels of increased risk were evident in majority-Black vs. white neighborhoods (42% increased risk) and CTs (66% increased risk). Finally, compared to high-income majority-white neighborhoods and CTs, there was a 39% increased risk of premature death in low-income majority-Black neighborhoods and a 63% increased risk in low-income majority-Black CTs (ICE$_{Race-Income}$). A poverty measure, estimating the proportion of the population below the federal poverty line, was also assessed. All three ICE measures outperformed the poverty measure at both geographical levels, with the steepest gradients detected at the CT level (see Table 2).

Krieger et al. [44] also conducted a study on child (< 5 years) and adult (< 65 years) premature chronic disease mortality in Massachusetts. Results indicated that low-income CTs had a 64% higher risk of child mortality than high-income CTs (ICE$_{Income}$). Similarly, in comparison to majority-white CTs, majority-Black CTs had an 85% higher risk of child mortality (ICE$_{Race}$), while low-income majority-Black CTs had a 119% higher risk than high-income majority-white CTs (ICE$_{Race-Income}$). The analogous risk ratios for adult premature mortality were considerably higher, ranging from 128%-139% increased risk (see Table 2). At the city/town level, results revealed a comparable but substantially weaker pattern (all RRs ≤ 1.40 for child mortality and ≤ 1.53 for adult mortality; see Table 2). When compared to a separate poverty measure, all three ICE measures accounted for more of the variance at the CT level. At the city/town level, however, the poverty measure accounted for slightly more of the variance in child (but not adult) mortality than each of the ICE measures (Table 2).

Another study by Krieger, Waterman et al. [28] examined the association between racialized economic segregation and both premature and diabetes-related mortality in New York

City at the CT and CD levels. They found that premature mortality risk was 124% higher in low- vs. high-income CTs ($ICE_{Income}$), 89% higher in majority-Black vs. majority-white CTs ($ICE_{Race}$), and 133% higher in low-income majority-Black vs. high-income majority-white CTs ($ICE_{Race-Income}$). The corresponding associations were even stronger for diabetes mortality risk ($ICE_{Income}$ = 185%; $ICE_{Race}$ = 178%; $ICE_{Race-Income}$ = 252%). Risks of either outcome at the CD level were comparable, though generally slightly weaker for premature mortality and slightly stronger for diabetes mortality (see Table 2). Consistent with their past studies, the authors employed an area-based poverty measure as an additional predictor. At the CT level, all three ICE measures detected stronger associations with each of the outcomes than did the poverty measure. At the CD level, however, the poverty measure accounted for more of the variance in diabetes mortality risk (249%) than $ICE_{Income}$ and $ICE_{Race}$ but less than $ICE_{Race-Income}$. For premature mortality, the poverty measure accounted for more of the variance (140%) than any of the ICE measures.

## Racialized economic segregation and other health outcomes

Two articles focused on SARS-CoV-2 outcomes. Brown, Lewis, and Davis [45] explored the link between racialized economic segregation and SARS-CoV-2 rates for incidence, testing, and positive test results for the calendar year of 2020. The study was conducted at the neighborhood level in Washington D.C. The results for the first six months showed significant negative correlations between incidence rates and $ICE_{Race}$, $ICE_{Income}$, and $ICE_{Race-Income}$. Similar significant negative correlations were evident for percent positive test results and each of the three ICE measures. By contrast, the ICE measures correlated positively with testing rates (see Table 2). Correlations for the second half of 2020 were comparable. These results show that while the most deprived neighborhoods exhibited the lowest testing rates, the incidence rates and percent positive test results were still higher in these neighborhoods than in the most privileged neighborhoods.

In the other study, Chen and Krieger [46] measured the national SARS-CoV-2 death rate at the county level. They also measured the Illinois rate of confirmed SARS-CoV-2 cases and the New York City rate of positive COVID-19 tests at the zip code tabulation area (ZCTA) level. Compared to high-income majority-white counties, low-income majority-Black counties had a 4% increased death rate ($ICE_{Race-Income}$) across the nation. In Illinois and New York City, low-income majority-Black as opposed to high-income majority-white ZCTAs had 219% increased rates of confirmed COVID-19 cases ($ICE_{Race-Income}$) and 68% increased rates of positive test results (rate ratios = 1.68), respectively.

Finally, Feldman, Waterman, Coull, and Krieger [47] found clear evidence of a positive association between racialized economic segregation and hypertension in racial and ethnic minority populations at the CT level. The odds of hypertension were 24% lower in majority-white vs. majority-Black CTs ($ICE_{Race}$). Similarly, odds of hypertension were 39% and 52% lower in high-income majority-white CTs as opposed to low-income CTs with high concentrations of people of color or Black residents, respectively ($ICE_{Race-Income}$).

## Discussion

### Main findings

The results reported in the reviewed papers support the conclusion that racialized economic segregation–as operationalized by the $ICE_{Race-Income}$ measure–is strongly associated with increased risk of a range of negative health outcomes in racial and ethnic minority populations. Indeed, all of the studies included in this review show significant links between $ICE_{Race-Income}$ and population health, including not only increased risk of PTB, infant mortality, and maternal death and morbidity, but also cancer, hypertension, COVID-19, and premature

mortality. Thus, based on the evidence, the health effects of racialized economic segregation may manifest in terms of increased risk of communicable diseases, chronic illnesses that develop over the life course, and even intergenerational health-risks and morbidities, transferred prenatally from mother to infant.

Our findings resonate strongly with those of other reviews on this topic which have reported similar connections between racialized segregation a broad range of health outcomes, including vascular diseases, cancer, pregnancy and birth complications, cancer, and lifestyle [9, 10, 48–51]. The present paper, however, advances current knowledge by highlighting poverty as a key pathway through which segregation produces negative health outcomes in racial and ethnic minority populations. Of the 18 studies that included statistics for all three ICE measures ($ICE_{Race}$, $ICE_{Income}$, and $ICE_{Race-Income}$), 67% found that combining racial/ethnic and economic segregation into a single variable ($ICE_{Race-Income}$) represented a better predictor of health than either racial/ethnic ($ICE_{Race}$) or economic segregation ($ICE_{Income}$) alone [28, 29, 31–33, 35, 38–41, 43, 44]. While effect sizes for $ICE_{Race}$ or $ICE_{Income}$ were slightly higher than $ICE_{Race-Income}$ in six studies, the difference was mostly negligible and often unstable (e.g., varying by geographical level) [24, 29, 31–33, 45]. Further, $ICE_{Race-Income}$ also outperformed other poverty measures applied to predict health outcomes in segregated populations, including the Hardship Index [35, 43] and other similar established measures of financial adversity [28, 31, 33, 42]. Altogether, the majority of the evidence suggests that conceptualizing residential segregation in multidimensional terms of overlapping concentrations of racial/ethnic minorities *and* poverty (rather than one or the other) may present a more complete and accurate account of this phenomenon and its relationship to health. We interpret these findings as reflecting a long and well-documented history of racialized segregation (e.g., through redlining policies) combined with persistent and systematic disinvestment and exclusion of racial and ethnic minority spaces and communities from the social and economic spheres of white society [2]. The evidence presented here thus illuminates the extent to which deep-seated structural drivers determine not only where minorities can live, but also the state of the physical space they inhabit, the accessibility of basic living necessities (e.g., health care, employment, education), and the opportunity for socioeconomic mobility. The critical assessment of the literature given in this review may thus provide impetus, as well as practical guidance, for the development and implementation of effective and incisive public health policies and interventions that target racialized economic segregation as a major structural determinant of social and racial inequalities in health.

## Themes and variations in the evidence

While the overall negative association between $ICE_{Race-Income}$ and minority health is consistent and clear across studies, several themes and variations were present in our review of the evidence. For example, in addition to income, a number of studies assessed other pathways that might further explain the link between racialized economic segregation and health. These include most prominently comorbidities and access to quality health care. Other themes in the reviewed evidence were of a more methodological nature and related to the performance of the ICE at different geographical levels and relative to traditional measures of disadvantage.

Two studies by Janevic et al. [34, 39] included the most extensive analysis of the pathways between racialized economic segregation and health. Their results demonstrated that the positive relationship between the ICE measures and both infant and maternal mortality and morbidity was mediated by the location of hospital of delivery. Specifically, women in low-income majority-Black areas were overwhelmingly more likely to give birth at local hospitals than at hospitals in more advantaged areas. This accounted for over one third of the observed

association between ICE$_{Race-Income}$ and risk of both maternal morbidity and adverse birth outcomes. The authors interpret this finding in terms of the generally decreased quality of care provided at hospitals in low-income majority-Black areas. Past research indicates that hospitals in deprived majority-Black areas face structural obstacles to their provision of quality care, including the capacity to attract highly trained staff and secure state-of-the-art equipment [52–55]. Further, from a patient perspective, individual-level socio-economic barriers (e.g., transportation, cost) may prevent individuals in these areas from accessing better care at non-local hospitals. Consistent with these points, while Brown et al. [45] did not test mediation, they argued that the insufficient COVID-19 testing resources allocated to low-income majority-Black neighborhoods might account for the observed lower testing rate and, by extension, the higher COVID-19 prevalence in these neighborhoods. In other words, the extent to which racialized economic segregation restricts access to quality health care and health promoting and prevention resources appears to account for the disproportionate negative health outcomes among racial and ethnic minority populations.

The increased prevalence of comorbidities among residents of deprived vs. privileged areas also featured in several studies as a mediator of the link between ICE$_{Race-Income}$ and specific health outcomes. Janevic et al. [39] found that pre-existing comorbidities partially explained the observed relationship between ICE$_{Race-Income}$ and SMM during pregnancy. Similarly, the results in Janevic et al. [34] suggest that the approximately seven-fold increased incidence of pre-pregnancy and gestational hypertension and diabetes in the most disadvantaged populations accounted for much of the correlation between ICE$_{Race-Income}$ and birth outcomes. Consistent with Janevic et al.'s studies, Dyer et al. [37] also found that pre-pregnancy diabetes and hypertension mediated the association between ICE$_{Race-Income}$ and pregnancy-related maternal mortality. Taken together, these studies indicate that exposure to racialized economic segregation over the life course may increase the likelihood of developing chronic health conditions which in turn exacerbate the risk of adverse pregnancy and birth outcomes. At a more general level, these results also signify the symbiotic connection between people and the spaces they occupy [8, 56]. That is, the most deprived areas typically comprise multiple elements in the built and social environments that inhibit healthy lifestyles, facilitate unhealthy behavior, and contribute to harmful living conditions. For example, past research has shown that disadvantaged areas are often disproportionately saturated with alcohol [57] and fast food outlets [58], have limited or non-existent exercise facilities [59] or green areas [60], and suffer from greater exposure to environmental pollutants (e.g., air pollution, hazardous housing materials, contaminated water supply) [27]. Other inhibitors of health might relate to socioeconomic pressure points that encroach on the individual's time, energy, sleep quality, and resources complicating any sustained pursuit of health. For example, residents in deprived areas may be more likely to be unemployed or work long hours at poorly paid jobs far from home, restricting their capacity and opportunity for self-care and healthy living [11]. In this way, if the physical and social characteristics of a given area comprise environmental health-risk factors and barriers to health behavior, it follows naturally that this will manifest in terms of greater vulnerability to disease and illness in the resident population.

Other themes in our synthesis of the literature relate to the implementation of the ICE$_{Race-Income}$ and its performance compared to other, more conventional measures of economic and racial segregation. As noted in the introduction, a key advantage of the ICE over other measures relates to its utility at a variety of geographical levels–particularly in smaller areas like blocks, neighborhoods, and CTs where traditional measures typically become impractical and uninformative [44]. This advantage was confirmed in our review. Across studies, the ICE was applied at six different geographical levels, including neighborhood, CT, CD, zip code, county, and city. In addition, four studies implemented the ICE at multiple levels for the purpose of

comparison, including zip code vs. county [46], neighborhood vs. CT [31], CT vs. CD [28], and CT vs. city [40]. In each spatial context, the ICE comprised an informative measure of segregation and its impact on health outcomes with comparable results at all levels of comparison. Further, six studies compared the statistical accuracy of the ICE to other traditional scales of disadvantage, including the Hardship Index [35, 43] and measures created from ACS data, reflecting the federal poverty line [28, 31, 42]. In nearly all of these comparisons, the ICE either matched or outperformed the other poverty measure (see Table 2). These findings attest to the versatile utility and application of the ICE as an effective tool for public health monitoring and research.

## Strengths, limitations, and future directions

A clear strength of the reviewed studies relates to their high methodological quality. As noted in the results section, none of the papers were of low quality and a large majority were of high quality ($n$ = 15). These ratings were mainly due to the rigorous research designs employed, the large sample sizes on which the studies were based, and the extensive efforts to control for relevant covariates. The extensive and high-grade secondary data sets (typically city, state, or national cohorts) leveraged in all of the studies also feeds into the overall merit of the evidence base.

The variety of outcomes across studies further strengthens the empirical credibility of the general association between segregation and health. The evidence links the ICE measures to not only chronic cardiometabolic illnesses like cancer, cardiovascular disease, and diabetes, but also communicable conditions such as COVID-19, and intergenerational health effects like adverse birth outcomes. Thus, exposure to racialized economic segregation appears to have pervasive and immediate as well as long-term negative health effects.

Another convincing feature of the evidence derives from the key focus of this paper. By employing the $ICE_{Race-Income}$ measure, the literature reviewed here provides a richer and more complete account of the multidimensional nature of segregation and its impact on population health. As noted, these studies advance the current understanding of residential segregation by revealing the extent to which structural forces concentrate populations into specific areas on the basis of race *and* income rather than one or the other. Given the polar directionality of the ICE, this measure also delineates the zero-sum nature of the relationship between deprived and advantaged populations, clearly demonstrating how concentrations of white privilege and wealth in one area results in concentrations of Black disadvantage and poverty in another. Acknowledging this underscores the problematic tendency in the literature to operationalize poverty and social adversity in unidimensional and unidirectional terms. Doing so appears to reflect an incomplete and misleading understanding of social disadvantage and privilege as somehow disparate and unrelated phenomena.

There are also limitations to the literature that need to be noted. First, the evidence base at large nearly exclusively comprises contrasts between Black and white populations. In fact, only three studies included other racial/ethnic populations. Westrick et al. [41] and Wiese et al. [42] both created ICE measures for both Black and white, and Hispanic and white contrasts, while Feldman et al. [47] contrasted white and 'people of color'. This homogeneity of the populations in focus represents a clear limitation to the generalizability of the current evidence base. However, the emphasis on Black-white contrasts might have its genesis in the unique history of Black-white segregation in the U.S. which has persisted since reconstruction [8]. Indeed, research indicates that high-income white and low-income Black populations continue to consistently occupy opposite ends of the socioeconomic continuum [8, 61, 62]. Given the extensive and increasing ethnic and racial diversity of the US population [63], future research should assess the link between segregation and health in other populations as well.

Second, another limitation to the generalizability of the reviewed results relates to the location of the studied populations. Only three studies were conducted in the South (Washington D.C., Louisiana, and Florida) while the rest were from Northern and Western states. As such, any conclusions drawn from the current evidence base only extend to these particular regions of the U.S.

Third, in terms of outcome variables studied, all pertained to physical health. No studies examined the effects of segregation on mental health. Pursuing this line of research might be exceedingly relevant as many of the effects of segregation–poverty, poor health care, employment, education, etc.–represent severe psychological stressors, which in all likelihood negatively affect mental health and well-being. In fact, extensive research indicates that the physiological reactivity associated with chronic psychological stress (i.e., allostatic load or "weathering") and poor mental health may over time result in physical illnesses, including cardiometabolic diseases and cancer [64–67]. This suggests that the observed physical health effects of segregation might also operate along psychophysiological pathways related to increased stress.

The risk of overcontrolling for covariates in statistical assessment models is a fourth notable limitation. Many of the studies in this review treated extraneous health-risk factors as confounders rather than potential pathways between the primary variables of interest. As such, these factors were typically entered into statistical models in blocks rather than in a stepwise sequence, preventing assessment of individual variable contribution to the overall model fit. However, as suggested in the studies by Janevic et al., Brown et al., and Dyer et al., failing to recognize and treat certain variables as potential pathways rather than covariates, risks oversimplifying rather than crystallizing the segregation-health relationship. This point dovetails with one of the primary messages of this review, which advances poverty as a central explanatory factor–rather than a covariate–of the link between segregation and health.

Extending on the previous point, a final limitation to the reviewed evidence base relates to the general lack of mediation analyses. Only four studies considered mediation and only two of these actually tested the mediating pathway between segregation and the outcomes [37, 39]. Past research suggests a multitude of factors that might further clarify how segregation produces the observed disparities in health. As previously mentioned, social ecological studies have shown dramatic differences between majority-white and racial/ethnic minority-concentrated areas on a range of variables in the social and built environment. These include food environment [58], exercise opportunities [59], alcohol outlet density [68], environmental pollution [27], crime [69], social cohesion [70], and access to green areas [60] and health care resources [4]–all of which may represent mechanisms through which structural racism produces population health disparities. In order to get a full picture of the segregation-health relationship, future research is needed to test process models that assess these and any other potential pathways.

## Practical and methodological implications

All strengths and limitations considered, the evidence presented in this review has several important practical and methodological implications. From an empirical perspective, the multidimensional format of $ICE_{Race-Income}$ provides added insight into the mechanisms that underpin the well-established link between residential segregation and health. Importantly, this opens up avenues for other variations of the ICE that might incorporate explanatory factors other than income. In fact, several studies have already tested new ICE measures that account for spatial polarization of deprivation and privilege by housing tenure (owners vs. renters) [30], education (high vs. low) [71, 72], as well as nativity (U.S. born vs. foreign) and

language (English vs. foreign language) [72]. While we find that ICE$_{Race-Income}$ accounts for a relatively large portion of the variance in the link between segregation and health, these new versions of the ICE may be useful in the effort to further disentangle this complex association and uncover additional underlying pathways. It is noteworthy that six studies in this review reported stronger associations between the outcome of interest and ICE$_{Race}$ and/or ICE$_{Income}$ compared to ICE$_{Race-Income}$. While these associations were inconsistent within and across studies and occurred in a minority of articles (30%), they may reflect the likelihood that there are other factors at play that interact with economic or racial segregation to impact on health. In line with this, we are currently exploring the possibility of creating another multidimensional ICE for the measurement of spatial concentrations of people who have been incarcerated–another racialized and key predictor of population health [73].

As noted in past research, and confirmed in our review of the evidence, the primary metric advantages of the ICE relate to its versatility in terms of application at different geographical levels and particularly in smaller areas like blocks, neighborhoods, and census tracts. It also offers intuitive insight into the directionality of the relationship in focus and is readily constructed using publicly available data (e.g., the ACS). Given these unique qualities, the ICE represents an exceedingly useful metric in social and health scientific research. In terms of 'real-world' application, the ICE is also well-suited as an effective public health monitoring tool. In this capacity, the ICE might be implemented to identify disadvantaged and privileged areas as well as the unequal distribution of resources that contribute to these disparities.

By advancing a more comprehensive understanding of the relationship between segregation and disparities in population health, the results of this review may serve to focus relevant policies and interventions that target social inequalities in health. For example, the studies by Janevic et al. and Dyer et al. identify hospital quality and comorbidities as potential points of intervention that could interrupt the connection between segregation and poor health outcomes. Similarly, constructing new ICE measures to identify additional pathways could further clarify where and how structural and/or social ecological changes can be achieved for optimal downstream effects on health equality.

## Conclusion

In the present paper, we critically review the empirical evidence as it relates to the link between segregation–as defined by ICERace-Income–and population-level health disparities. We find that exposure to racialized economic segregation significantly increases the risks of a range of negative health outcomes, including PTB, infant mortality, cardiometabolic disease, and all-cause mortality. First and foremost, these findings underscore the dire health effects that follow in the wake of the structural forces that protect and perpetuate the privilege conferred on majority-white populations by pushing racial and ethnic minorities into chronically underserved and low-income geographical spaces. Our assessment of the evidence also highlights the utility of the ICE measure in social and health scientific research, providing a clear avenue for future study to further advance the knowledge on the mechanisms by which racial segregation creates health disparities. At a more applied level, the evidence also emphasizes the potential of the ICE as an effective public health monitoring tool that may serve to identify at-risk areas, determine optimal points of intervention, and inform top-down policies that target racialized economic segregation and the associated health disparities.

## Supporting information

**S1 Checklist. PRISMA 2009 checklist.**
(DOC)

## Acknowledgments

We are grateful to Elvis Alexis Maliza and Mikael Polanco for their assistance in the literature search results evaluation and data extraction.

## Author Contributions

**Conceptualization:** Anders Larrabee Sonderlund, Mia Charifson, Antoinette Schoenthaler, Traci Carson, Natasha J. Williams.

**Data curation:** Anders Larrabee Sonderlund, Mia Charifson.

**Formal analysis:** Anders Larrabee Sonderlund, Mia Charifson, Antoinette Schoenthaler, Natasha J. Williams.

**Methodology:** Anders Larrabee Sonderlund, Mia Charifson, Antoinette Schoenthaler, Traci Carson, Natasha J. Williams.

**Supervision:** Antoinette Schoenthaler, Natasha J. Williams.

**Visualization:** Anders Larrabee Sonderlund, Mia Charifson, Antoinette Schoenthaler, Traci Carson, Natasha J. Williams.

**Writing – original draft:** Anders Larrabee Sonderlund, Mia Charifson.

**Writing – review & editing:** Anders Larrabee Sonderlund, Mia Charifson, Antoinette Schoenthaler, Traci Carson, Natasha J. Williams.

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
