## [Editor Report · Decision Letter 0]

18 Oct 2021

PONE-D-21-31241Racialized economic segregation and health outcomes: A systematic review of studies using the Index of Concentration at the Extremes of race, income, and their interactionPLOS ONE

Dear Dr. Larrabee Sonderlund,

Thank you for submitting your manuscript to PLOS ONE. After careful consideration, we feel that it has merit but does not fully meet PLOS ONE’s publication criteria as it currently stands. Therefore, we invite you to submit a revised version of the manuscript that addresses the points raised during the review process.

We look forward to receiving your revised manuscript.

Kind regards,

Tariq Jamal Siddiqi

Academic Editor

PLOS ONE

Journal Requirements:

Additional Editor Comments:

Sonderlund et al. has performed a systematic review, “Racialized economic segregation and health outcomes: A systematic review of studies using the Index of Concentration at the Extremes of race, income, and their interaction” in which they have shown that racialized economic segregation is strongly associated with increased risk of a range of negative health outcomes in racial and ethnic minority populations. In my opinion, this article can be improved by incorporating the following points:

1. The authors did not give p-values for the findings they obtained when addressing ethnic groups in the results.

2. When addressing ethnic differences and results which are significant or non-significant, adding the effect sizes such as the odds ratio or risk ratios found in the review, may improve the findings even further.

3. Separate tables to summarize results or making sections in the same tables for the different outcomes such as cancer, preterm birth or covid-19 etcetera, can enhance the findings of the review further.

4. A few statements about which financial adversity measures does the ICE (Race-Income) measure outperforms can contribute to enhance that point of discussion.

5. More points can be added to the discussion regarding the findings of other studies related to the outcomes assessed in this review like the results in their outcomes that are relevant to the present research to enhance the discussion.

6. When discussing barriers to improved healthcare in underserved regions, it may be helpful if the writers provide evidence of poor healthcare practice in the specified places.
---

## [Author Response · Author response to Decision Letter 0]

28 Oct 2021

Dear Editor,

Thank you for your helpful comments and suggestions for improving our manuscript. Below we have gone through each of the points you raised and revised our paper accordingly. We feel that it has improved significantly as a result.

Kind regards,

Anders Larrabee Sonderlund, Mia Charifson, Antoinette Schoenthaler, Traci Carson, and Natasha J. Williams.

1. The authors did not give p-values for the findings they obtained when addressing ethnic groups in the results.

• We have revised Table 2 and added p-values from those papers that reported them. 

2. When addressing ethnic differences and results which are significant or non-significant, adding the effect sizes such as the odds ratio or risk ratios found in the review, may improve the findings even further.

• We confirm that all effects sizes found in the review are reported in Table 2. We considered including these statistics in the body text instead, but ultimately decided that a table would provide a better overview of effect sizes reported in each paper.

3. Separate tables to summarize results or making sections in the same tables for the different outcomes such as cancer, preterm birth or covid-19 etcetera, can enhance the findings of the review further.

• Agreed. We have reorganized Table 2 and grouped results by outcome to reflect the four subsections of the Results section. 

4. A few statements about which financial adversity measures does the ICE (Race-Income) measure outperforms can contribute to enhance that point of discussion.

• This is a good point. The reviewed studies compared the ICE to either the Hardship Index or poverty measures constructed based on American Community Survey data. We have revised parts of the discussion to properly clarify this:

Page 27, lines 91-94: “Further, six studies compared the statistical accuracy of the ICE to other traditional scales of disadvantage, including the Hardship Index (35, 43) and measures created from ACS data, reflecting the federal poverty line (28, 31, 33, 42). In nearly all of these comparisons, the ICE either matched or outperformed the other poverty measure (see Table 2).”

5. More points can be added to the discussion regarding the findings of other studies related to the outcomes assessed in this review like the results in their outcomes that are relevant to the present research to enhance the discussion.

• We thank he editor for this suggestion. While the extant research relevant to this review is detailed to some extent in the introduction as well as the discussion (e.g., strengths and limitations section), we have attempted to accentuate these points further. Specifically, we have revised the second paragraph (Page 24, lines 11-14) of the Discussion section to emphasize the broader empirical context within which our review sits. 

6. When discussing barriers to improved healthcare in underserved regions, it may be helpful if the writers provide evidence of poor healthcare practice in the specified places.

• Agreed. We have revised the text on pp. 25-26 and added relevant references to this point:

Page 25, lines 49-51: “Past research indicates that hospitals in deprived areas face structural obstacles to their provision of quality care, including the capacity to attract highly trained staff and secure state-of-the-art equipment (52-55).”

Page 26, lines 64-65: “In other words, the extent to which racialized economic segregation restricts access to quality health care and health promoting and prevention resources appears to account for the disproportionate negative health outcomes among racial and ethnic minority populations.”

---

## [Decision Letter · Decision Letter 1]

10 Jan 2022

Racialized economic segregation and health outcomes: A systematic review of studies using the Index of Concentration at the Extremes of race, income, and their interaction

PONE-D-21-31241R1

Dear Dr. Larrabee Sonderlund,

We’re pleased to inform you that your manuscript has been judged scientifically suitable for publication and will be formally accepted for publication once it meets all outstanding technical requirements.

Kind regards,

Tariq Jamal Siddiqi

Academic Editor

PLOS ONE
---

## [Editor Report · Acceptance letter]

20 Jan 2022

PONE-D-21-31241R1 

Racialized economic segregation and health outcomes: A systematic review of studies using the Index of Concentration at the Extremes of race, income, and their interaction 

Dear Dr. Larrabee Sonderlund:

I'm pleased to inform you that your manuscript has been deemed suitable for publication in PLOS ONE. Congratulations! Your manuscript is now with our production department. 

Kind regards, 

on behalf of

Dr. Tariq Jamal Siddiqi 

Academic Editor

PLOS ONE